# Implementation of the New European Bauhaus Principles as a Context for Teaching Sustainable Architecture

Kajetan Sadowski 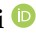

Faculty of Architecture, Wrocław University of Science and Technology, 50-317 Wrocław, Poland; kajetan.sadowski@pwr.edu.pl

**Abstract:** Due to the presentation of the European Green Deal (EGD) on 11 December 2019, it is important to introduce a new context for the education of architects corresponding to the objectives set by the European Union. These include reducing greenhouse gas emissions, increasing the energy efficiency of buildings, designing buildings in accordance with the principles of the circular economy, using renewable energy, as well as promoting ecological food and protecting biodiversity. As part of the design course *Environmentally Friendly Housing Architecture*, inspired by, among others, the design of the New European Bauhaus and the former Bauhaus art school, both of which are compared in the first part of this article, we identify a number of new, assessed design indicators related to the achievement of the above objectives, in line with the trend of sustainable architecture. The indicators are divided into four main categories: energy, environment, indoor climate, and society, where, for example, the environmental category includes the following criteria: embodied energy ($MJ/m^2$), embodied carbon footprint ($CO_2eq/m^2$), use of rainwater and gray water (% of demand), use of mains water (% of demand), local production of vegetables and fruit (% of demand). During the design process, changes were made to achieve better indicators, and the final designs were described using radar charts. The paper presents a statistical summary of the achieved values for individual indicators, the progress achieved, exemplary design solutions, and the assessment of the methodology used. The design course *Environmentally Friendly Housing Architecture* was assessed by the participants by means of a questionnaire.

**Keywords:** sustainable design; Bauhaus; New European Bauhaus; renovation wave; architectural education; energy efficiency; carbon footprint

## 1. Introduction

On 16 September 2020, the new president of the European Commission, Ursula von der Leyen, made a speech on the state of the Union in which an ecological, economic, and cultural project called the New European Bauhaus (NEB) was announced [1]. It is an initiative that combines design, ecology, social, and price accessibility aspects as well as investments in order to support the implementation of the European Green Deal (EGD) [2]. The name, referring to the Bauhaus—the German arts and crafts school from the first half of the 20th century, signals a common source of inspiration, which is the need to develop in the face of social and technological changes taking place in the world. The recently celebrated 100th anniversary of the founding of the Bauhaus was, therefore, an excellent excuse to re-look at its heritage, especially in the context of a holistic approach to design. The beginning of the return to the Bauhaus is the idea of the Baukultur movement, developed for over 20 years and formally initiated in Germany as the Federal Baukultur Foundation. It aims to combine a high standard of design with a holistic view of social, economic, environmental, and cultural aspects. Creating Baukultur is a social process based on a broad understanding of values and goals and high-quality interdisciplinary discourse [3]. The signing of the "Declaration Towards a High-Quality Baukultur for Europe" in 2018 in Davos, the inclusion of Baukultur in the work of the European Council,

the elections to the European Parliament, which were dominated by problems related to the climate crisis and the development of the poorer parts of Europe, and, eventually, the COVID-19 pandemic have shifted the center of gravity of the upcoming changes to connect Baukultur with the challenges of climate protection and the recovery of the economy after the pandemic, which may be an opportunity for a fresh start.

The first part of the "Sixth IPCC Report", published on 9 August 2021, unquestionably states that global warming is a result of human activity and recalls the conclusions of the previous report on the need to reduce $CO_2$ emissions to a net-zero level [4]. Since construction, as part of the economy, is responsible for a significant part of these emissions (38%) [5], the success of achieving this goal depends on actions in this area.

In this context, the activities of the European Commission are up-to-date, and the goal of achieving climate neutrality by 2050, presented in the EGD, is ambitious on a global scale. The founder of the Bauhaus, Walter Gropius, argued that the projects should enable the expression of thoughts and moods of their time, which, at the beginning of the 20th century was, for example, the growing demand for inexpensive and healthy housing, to which prefabrication was supposed to be the answer [6]. In the case of the New European Bauhaus, the problem is the climate crisis, and the answer to how to solve it is still sought; hence, the NEB is to be a platform for the development of new ideas. It is to create a bridge between the world of art and culture on one hand and the world of science and technology on the other; this will involve the whole of society: artists, students, architects, engineers, scientists, and innovators. It is supposed to be a system change [1]. Additionally, since every change requires education, teaching environments need to respond to new challenges and integrate curricula with the requirements of the upcoming change.

## 2. Bauhaus Ideas as a Response to the Challenges of the Early Twentieth Century

The name Bauhaus, founded in 1919 in Weimar by Walter Gropius, an art and craft school, refers to the medieval organization *Bauhütte*, associating bricklayers, stonemasons, carpenters, and other craftsmen working on large construction sites. It was not a coincidence because the Bauhaus was supposed to be a return to artisanal tradition in the spirit of the Art And Crafts movement founded in 1988 in England, promoting the creation of art both aesthetic and useful, but without its opposition to industrial production. On the contrary, Bauhaus wanted to combine the achievements of modern technology with art, thus creating a concept of modern industrial design.

Despite the short period of activity (1919–1933), the school developed a mature educational program, the aim of which was the ability to holistically design objects or buildings that meet the needs of a changing society, taking into account the technological development of industry and, at the same time, durable utility and aesthetic values. To implement this modern approach to design, a modern teaching system based on outstanding designers and a collective sense of common purpose was also needed. The school functioned for only a dozen years, but it was created, apart from Walter Gropius, by outstanding lecturers, including Paul Klee, Wassily Kandinsky, László Moholy-Nagy, Marcel Breuer, and Ludwig Mies van der Rohe, who was the last of its directors. The program had many consistent features, e.g., parallel theoretical and practical learning of crafts; interdisciplinary learning, with many lecturers working on one project; working together in groups and even living together [7]; use and synergy of many additional arts and cultural elements such as dance, poetry, music, costumes, leading to a sense of inner bonds, solidarity, and a sense of mission and responsibility among the participants. Thanks to this, the ideas of the Bauhaus were durable and could develop effectively, transferred by its members to other countries, even after the school was closed by the Nazi authorities in 1934. The ideological and design achievements of the Bauhaus, despite its short period of operation, had a huge impact on the development of the art of design, bringing about various projects (Figure 1), ranging from simple household appliances (Wilhelm Wagenfeld lamps (1923)) and furniture (e.g., Wassily (1925–1926) and Cesca (1928) chairs by Marcel Breuer, the Barcelona armchair (1929) by Mies van der Rohe) to buildings (ending with the headquarters of the school

in Dessau (1926, designed by Walter Gropius), house Haus am Horn in Dessau (1923, designed by Georg Muche), and the Barcelona Pavilion (1929, designed by Mies van der Rohe)).

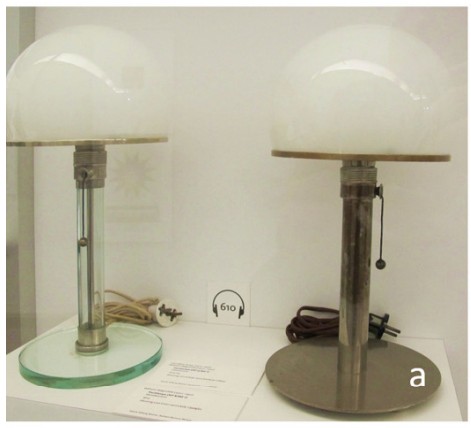
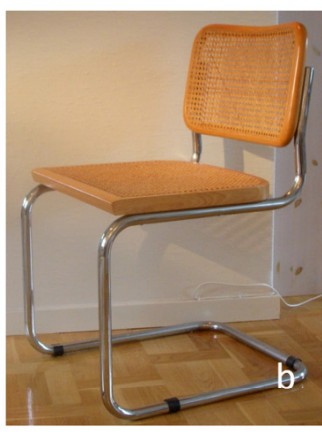
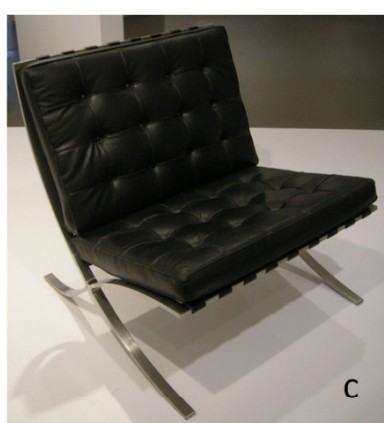
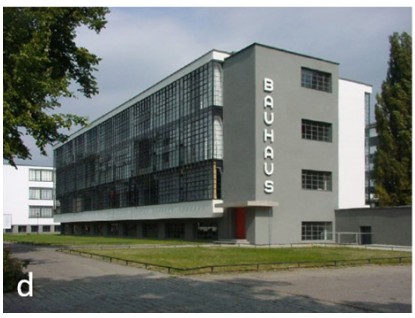
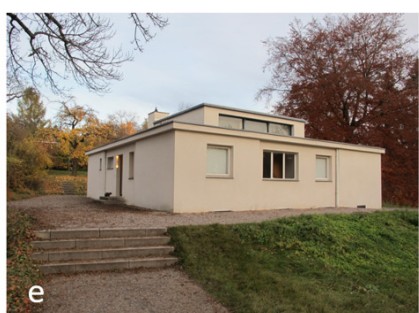
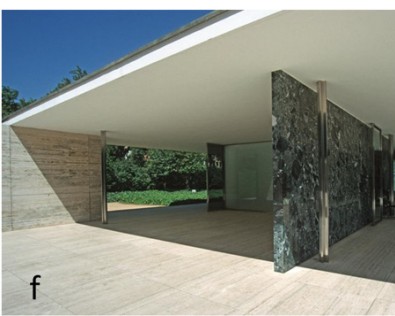

**Figure 1.** Examples of *Bauhaus* designs: (**a**) Wagenfeld lamp (photo by I. Sailko, CC BY-SA 3.0, https://commons.wikimedia. org/w/index.php?curid=17436047, accessed on 24 August 2021); (**b**) Cesca chair (photo By Holger Ellgaard—own work, CC BY-SA 3.0, https://commons.wikimedia.org/w/index.php?curid=4575023, accessed on 24 August 2021); (**c**) Barcelona chair (photo by I. Sailko, CC BY 2.5, https://commons.wikimedia.org/w/index.php?curid=6334832, accessed on 24 August 2021); (**d**) Dessau school building (photo by Mewes, public domain, via Wikimedia Commons https://commons.wikimedia.org/ w/index.php?curid=165180, accessed on 24 August 2021); (**e**) Haus am Horn House (photo by I. Sailko, CC BY-SA 3.0, https: //commons.wikimedia.org/w/index.php?curid=17433782, accessed on 24 August 2021); (**f**) Barcelona Pavillion (photo by Hans Peter Schaefer, CC BY-SA 3.0, via Wikimedia Commons https://commons.wikimedia.org/w/index.php?curid=52728, accessed on 24 August 2021).

The idea of universal education consisting of the gradual acquisition of skills—from general to specific, including teaching history only at the final stage of studies so as not to discourage and not to restrict the ideas of younger students, made it possible to acquire universal skills in solving design problems in all areas. This gave rise to the development of modern design, which is ubiquitous today. These days, people trained in design have gained access to a very wide range of professions, and, together, they exert a tremendous influence in positions of senior management, government, and academia. It is a testimony to the quality of design practices and the importance of design education in contemporary society [8].

## 3. Challenges of the New European Bauhaus in the Context of the *Bauhaus* School

One of the tools supporting the achievement of EGD goals is the ecological, economic, and cultural project of the New European Bauhaus (NEB), the design phase of which was launched in January 2021. Its most important values are sustainable development, aesthetics, and social participation, and the aim of the design phase (first half of 2021) is to search and define the final shape of the initiative, which will be implemented in five European countries in subsequent phases and, finally, promoted in the EU and beyond



its borders. The NEB is to initiate a new approach to the most important challenge of our time—the climate crisis. This approach is to be modeled on the principles used in the Bauhaus school, as shown in the comparison in Table 1:

**Table 1.** Comparison of the features of the Bauhaus school with the goals of the NEB.

| Basic Principles Introduced by the Bauhaus | Principles Promoted by the NEB |
|---|---|
| Form follows function: According to this idea, simple but elegant geometric shapes were designed based, in part, on society and did not respond to the intended function or purpose of a building or an object [9]. This postulate rose in contrast to the ornamentation widely used in architecture, which satisfied the needs of the varied problems related to rapidly growing populations and their housing needs. | NEB is an invitation to change perspectives and look at our green and digital challenges as opportunities to transform our lives for the better [2]. NEB responds to the most important challenge of our time—the climate crisis—and the efforts it undertakes to create a new, sustainable society. |
| No border between artist and craftsman: In a pamphlet for an April 1919 exhibition, Gropius stated that his goal was to create a new guild of craftsmen without class distinctions, which had raised an arrogant barrier between craftsman and artist [10]. The principle of combining various specializations while working on one project was also applied [11]. This was also the purpose of the parallel theoretical and practical work [6]. | Ursula von der Leyen, in her speech on the state of the European Union, emphasized that "we need to give our systemic change its own distinct aesthetic—to match style with sustainability. This is why we will set up a new European Bauhaus—a co-creation space where architects, artists, students, engineers, and designers work together to make that happen" [1]. This creative work, carried out on the basis of the collaboration of many specialists, corresponds to the new kind of artisans promoted by the Bauhaus. |
| True materials and smart use of resources: Materials should reflect the true nature of objects and buildings. Bauhaus architects did not hide even brutal and rough materials [9]. It is all about the smart use of resources, with a zero-waste ideal in mind [12]. | NEB aims to mobilize designers, architects, engineers, scientists, students, and creative minds in various disciplines to take a new look at sustainable lifestyles in Europe and the world [2]. This view is to take into account the current challenges of the European Union—energy neutrality by 2050, creating a sustainable society and circular economy. |
| Emphasis on technology: Bauhaus workshops were used for developing prototypes of products for mass production. The artists embraced the new possibilities of modern technologies [9]. | NEB aims to be a fresh approach to developing innovative solutions to complex social problems through co-creation. It is also a bridge between the world of science and technology and the world of culture and art [2]. |
| Simplicity and Effectiveness: There is no need for additional ornamenting and to make things more and more 'beautiful'. They are just fine as they are [9]. Bauhaus put functionalist design first and foremost to meet the needs of users. The effect was supposed to be useful, cheap, but also aesthetic. | NEB will propose and co-finance all innovative solutions. Efficiency will have a special place in the package of expected solutions, in this case, energy and the circular economy, which is one of the main goals of the EGD and the Renovation Wave project [1]. |
| Constant development: Bauhaus is all about new techniques, new materials, new ways of construction, and new attitudes, all the time. Architects, designers, and artists have to invent something new all the time [9,12]. | NEB is to be a platform for experimentation and communication, facilitating the cooperation of creators who want to design future conditions and lifestyles [2]. Continuous experimentation means continuous development, constantly meeting the changing needs of society. |

Therefore, it is necessary to emphasize the great ideological similarity between the NEB and Bauhaus. Today, after 100 years, the challenges facing society have changed, but the way of solving them is universal and, most importantly, holistic, based on the cooperation of specialists in many fields to re-shape the technology, economy, and culture of Europe in line with the objectives of the EGD.

## 4. Objectives of the European Green Deal in the Context of Sustainable Design

The European Commission's legislative initiative, the European Green Deal (EGD), aims to adapt the EU's climate, energy, transport, and tax policies to meet the goal of reducing net greenhouse gas emissions by 2030 by at least 55% compared to 1990 levels. This very close time horizon requires European countries to take decisive measures, which in the field of construction, according to the European Commission, are primarily the decar-

bonization of the construction sector, the renovation of existing housing stocks, increasing the energy efficiency of buildings (reduction of final and primary energy consumption by 36–39%), and increasing the share of renewable energy sources (up to 40%) [13]. In addition, on 14 July 2021, the European Commission adopted a package of proposals called Fit for 55, summarizing the above projects and giving them an even more specific legal framework. In addition to the above-mentioned goals for 2030, new goals [14] were presented, including:

- the share of renewable energy sources in buildings should be at least 49%;
- increasing the use of energy from renewable sources in heating and cooling by 1.1% per year;
- increasing the use of renewable energy in district heating and cooling systems by 2.1% per year;
- reduction of energy consumption in the public sector by 1.7% per year;
- reduction of primary energy consumption by 39%;
- reduction of final energy consumption by 36%;
- renovation of 3% of public buildings annually.

The European Commission has divided the challenges of the EGD into several areas of action: Climate, Energy, Agriculture, Industry, Environment and Oceans, Transport, Finance, and Regional Development, Research, and innovation. Achieving these goals will require European countries to take decisive actions for which specialists will be needed; some of them will cover areas that are the subject of architectural activities. The European Commission estimates that 160,000 new jobs will be created in the construction sector alone to meet the building modernization target [13]. This requires considerable effort to educate future architects as well as academic staff. As research shows, there is a need to educate academic staff for sustainability leadership as well as to update curricula with the simultaneous definition of the concept of sustainability. It is pointed out that the principles of Education for Sustainability (EFS, [15]) are still far from being integrated into the everyday practices of academic staff, and one of the main reasons for this is the complexity, ambiguity, and multidimensionality of the concept of sustainable development, which makes it difficult for individuals to understand why and how to implement sustainable development in practice or how to educate for sustainable development [16].

In the case of the author's study, it was decided to define the concept of sustainable design in relation to the goals presented in the EGD initiative. The following areas and goals of initiatives have been identified in relation to the features of sustainable design to achieve them:

The summary indicated in Table 2 shows that the objectives and areas of activities of the EGD and the NEB supporting it are consistent with the principles of sustainable design. On this basis, a method of teaching architecture students was proposed, and then the effectiveness of its application was tested in the context of sustainable design skills and the impact of the method used on the sense of responsibility for the actions taken.

**Table 2.** Selected architectural activities in relation to the areas of activities to implement the EGD (from A to E) to the features of sustainable architecture (from I to XXVII).

| Action Areas to Implement EGD (from A to E) in Relation to Architectural Design | Features of Sustainable Architecture (from I to XXVII) |
|---|---|
| (A) Selected targets in the area of Climate [17]: <br> - Net-zero greenhouse gas (GHG) emissions by 2050 <br> - Reduction of GHG emissions to 55% by 2030 compared to 1990. | - (I) Use of energy-saving solutions [18,19] <br> - (II) Use of renewable energy sources [20,21] |

**Table 2.** *Cont.*

| Action Areas to Implement EGD (from A to E) in Relation to Architectural Design | Features of Sustainable Architecture (from I to XXVII) |
|---|---|
| (B) Selected goals in the area of Energy [22]: <br>- Reaching the level of 32% (possibly 40%) of energy from renewable sources by 2030. <br>- Reaching 32.5% energy efficiency savings in terms of final and primary energy consumption by 2030 (possibly 36% and 39%, respectively) compared to the 2007 projections. <br>- Prioritizing energy efficiency, improving the energy performance of buildings, and developing an energy sector based mainly on renewable sources. <br>- To fight against energy poverty. <br>- Renovation of at least 3% of the total floor area of all public buildings per year. <br>- Increasing the use of renewable energy for heating and cooling buildings by 1.1% per year by 2030. | - (III) Taking into account the shape, surroundings, and orientation of the building [18,19,23–25]. <br>- (IV) Designing the optimal form and building envelope [19,24,25]. <br>- (V) Use of energy-saving solutions for heating, cooling, hot water, lighting, and ventilation [18,19]. <br>- (VI) Taking into account the environment in design, construction, and use [18,19,26,27]. <br>- (VII) Taking into account the quality of life of residents during design, construction, and use [26,28,29]. <br>- (VIII) Taking into account adaptation to the changing environment [18,30]. <br>- (IX) Use of renewable energy sources [20,21]. |
| (C) Selected objectives in the area of Agriculture [31,32]: <br>- Reduction of the environmental and climate footprints related to the food system. <br>- Leading global transformation towards competitive sustainability *farm to fork*. <br>- Promoting organic farming. <br>- At least 25% of the EU's agricultural land dedicated to organic farming and a significant increase in organic aquaculture by 2030. <br>- Strengthening local and low-value processing and promoting short trade. <br>- Reduction of climate and environmental footprints | - (X) Designing biologically active areas [27,33]. <br>- (XI) Enabling local food production [34–36]. <br>- (XII) Restriction of the transport of food products [34–36]. <br>- (XIII) Supporting biodiversity [15]. <br>- (XIV) Respect for the surrounding environment [18,19,26,27]. |
| (D) Selected objectives in the area of Environment and Oceans [37–40]: <br>- Transition to a circular economy. <br>- Recycling of waste. <br>- From farm to fork strategy. <br>- Forest strategy, including planting 3 billion additional trees by 2030. <br>- Supporting the production of sustainable food. | - (XV) The use of natural materials [37]. <br>- (XVI) Environmentally friendly production of materials [41]. <br>- (XVII) Use of local materials [41]. <br>- (XVIII) Use of recycled materials [19,30,41]. <br>- (XIX) Use of an effective, easy-to-assemble structure [18,19]. <br>- (XX) Use of materials with zero and reduced carbon footprints [42,43]. <br>- (XXI) Taking into account the circular economy, reusable materials [10]. <br>- (XXII) Reduction of the amount of waste [41–43]. <br>- (XXIII) Reduction of water consumption and the effective use of rainwater [18,19]. |
| (E) Selected objectives in the area of Transport [44]: <br>- Reduction by 55% of emissions from passenger cars by 2030 <br>- Reduction by 50% of emissions from vans by 2030. <br>- Zero emissions from new passenger cars by 2035. <br>- 90% reduction in transport-related greenhouse gas emissions by 2050. | - (XXIV) Designing in a way that reduces the need for communication [45–47]. <br>- (XXV) Limiting the need to use road transport [45–47]. <br>- (XXVI) Facilitating walking and cycling [46,47]. <br>- (XXVII) Facilities for low-emission vehicles. |

## 5. Aims and Scope

The article presents the methodology and results of the implementation of NEB assumptions and the related principles of sustainable design as part of the *Environmentally Friendly Dwelling Architecture* course implemented at the second degree of studies at the Faculty of Architecture, Wrocław University of Science and Technology. The aim of the

research is to demonstrate the possibility of implementing the EGD and NEB principles using the methods of sustainable design, the impact of their application on design skills, taking into account the challenges of the climate crisis, the impact on the increase in environmental awareness, and the assessment of the results and usefulness of such an approach in the educational process. The article indicates the following principles and methods of education developed on the basis of the NEB guidelines and the theoretical and practical knowledge of the author:

- educational principle of four-stage works on the project;
- principles of sustainable architecture;
- principles of education referring to the principles implemented at the Bauhaus art school;
- principles deriving from the objectives of the EGD;
- a method of assessing the effectiveness and purposefulness of research;
- the survey method as a didactic and scientific experiment.

The course was carried out in the academic year 2020/2021; it was an elective course, and 15 students (divided into 6 project groups) took part in it.

## 6. Teaching Method and Scope of the Course

The subject of the course was the design of multi-family residential buildings located in Wakefield, Ontario, Canada, that meet the requirements of sustainable and environmentally friendly construction. There were several plots to choose from, divided into two groups—urban plots and suburban plots. The guidelines for each plot type were as follows:

Urban plots:

The designed space should consist of mutually interpenetrating neighbor spaces with cheap, eco-architecture, made using environmentally friendly technology (e.g., modular).

As part of the architectural concept, the following spaces had to be designed:

- residential—multi-family, with a diversified structure of apartments: M1 (36–46 m$^2$), M2 (72–82 m$^2$), M3 (108–118 m$^2$), M4 (120–130 m$^2$), and M5 (144–154 m$^2$), connected with outdoor, open, or built-up individual or community spaces, with assumed specificity;
- ecological cultivation (permaculture gardens), use and recovery of rainwater, and hobby and service spaces (technical, utility rooms, waste);
- pro-social/community/co-working (e.g., shared kitchens, small gastronomy, studios, multi-functional rooms, community gardens);
- eco-educational (e.g., sensory gardens, permaculture crops);
- recreational (e.g., yoga, fitness, open sports areas);
- office and service (e.g., workplaces, shops);
- Service area (utility room, technical room).

The proposed assumption should include solutions ensuring active and passive energy gains, the largest possible biologically active surface, and the recovery and use of rainwater.

Suburban plots:

The designed space should consist of mutually interpenetrating, complementary spaces:

- experimental, residential: single-family and low-rise multi-family (including intermediate city villas);
- a diversified structure of apartments: M1 (36–46 m$^2$), M2 (72–82 m$^2$), M3 (108–118 m$^2$), M4 (120–130 m$^2$), and M5 (144–154 m$^2$), connected with outdoor, open, or built-up individual or community spaces, with designed specificities: ecological cultivation (permaculture gardens), use and recovery of rainwater, and service spaces (technical room, utility room, waste);
- pro-social/community/co-working (e.g., shared kitchens, small gastronomy, studios, multi-functional rooms, community gardens);
- eco-educational (e.g., forest kindergarten, ecological education center, sensory exhibition space, ecological education park—closed, and a local wild ecosystem, created on the basis of the existing forest and park space, sensory gardens, permaculture crops);
- recreational (e.g., yoga, fitness, outdoor sports areas);

- service area (utility room, technical room).

Other guidelines:

The planned assumptions on both urban and suburban plots were to take into account the following requirements and optional pro-ecological solutions:

- respecting and referring to the local biosphere in the design process;
- preferred use of local, natural materials, taking into account the life cycle of the materials;
- inclusion at the pre-project analysis stage of energy simulation on the urban model, water circulation and retention, zone and sector analysis, and a planting plan;
- conscious design of facilities, taking into account their energy balance and carbon footprint;
- conscious design of the building surroundings, taking into account the area for vegetable gardens and/or box crops;
- conscious design with the choice of renewable energy sources;
- conscious design of retention pools, rainwater collection sites;
- conscious design of composters, waste segregation, and recycling sites;
- conscious design of eco-transport (bicycle shops, carsharing, and electric car charging stations);
- conscious design of permaculture gardens, hives, optional mini-farms, and farms (along with the rooms necessary to handle animals).

Work was carried out in groups of 2–3 students, thanks to which the learning was carried out in a more effective way [48,49], in line with the rules applied in the Bauhaus school [7].

Another principle applied in accordance with the demands of Baukultur, the Bauhaus, and the NEB was the cooperation of students with specialists in many fields. During the course, individual classes in the following weeks were divided into two consecutive parts—the theoretical part, led by a specialist in a given field, and the practical part, during which consultations took place (usually with the specialist from the previous week). Thanks to this, the principles of collective cooperation of many specialists on one topic, combining the theoretical and practical parts [1,6,11], were implemented. The following lectures were carried out, combined with consultations conducted by nine specialists (in brackets are indicated the areas of EGD activities from A to E (Table 2) and the features of sustainable architecture from I to XXVII (Table 2) carried out by a given lecture):

1. Energy modeling of the building using SketchUp Pro 2021 and Sefaira software, v. 3.0.0 (A, B/I, III, IV, V).
2. Rainwater retention installations (D/XXI, XXIII).
3. Home crops (B, C/VI, X-XIV).
4. Eco-design (A-E/I-XXVII).
5. Eco-housing architecture integrated with ecological education parks in the suburban environment (C, D/X, XIII, XIV, XVI, XXII, XXIII).
6. Innovative modular construction (B, D/VI, XIX).
7. LCA analysis using Athena Impact Estimator software, v. 5.4 (D/XV, XVI, XVII, XVIII, XX).
8. Universal design (B/VII, VIII).
9. Greenery as an element of environmentally friendly architecture (B, C, E/VI, X-XIV, XXIII).
10. Photovoltaic installations (A, B/II, V, IX).
11. CLT wood constructions (D/XVI, XX).
12. Modern building structures (D/XIX).
13. Futuristic architecture (B/VII, VIII).

Work on the project was carried out in four stages (Table 3):

**Table 3.** Evaluation of the stages of work on the design concept.

| Stage | The Scope and Method of Works Performed | Type of Drawings Performed | Subject and Purposefulness of the Research, Method |
|---|---|---|---|
| Stage 1 | Stage 1 is devoted to a deep inventory of the area consisting of collecting photographic documentation, underlays in the form of maps, and performing analyses of the views, communication, green, functional, sunlight, rainfall and snowfall, temperature, winds, acoustics, and features of the surrounding buildings. | Any form | The work was of an analytical nature, during which the collected data was processed and conclusions were described. The work provided the basis for the development of design guidelines in the context of optimal land use in terms of sustainable design. Analysis and synthesis method: This stage ended with a discussion and evaluation. |
| Stage 2 | Stage 2 consists of the development of an urban concept, including a planned development complex with the immediate surroundings, taking into account the functional, spatial and communication structure. | Stage 1 range and additionally: <br> - idea presentation; <br> - urban concept 1:1000; <br> - fragment of the master plan 1:500; <br> - simplified 3D model; <br> - energy simulation. | The work consisted of design in which several variants were presented and then discussed. The selected variant takes into account the environment, communication, acoustic, sunlight, shade, winds, and energy efficiency (performed in Sefaira software). Analysis and synthesis method: This stage ended with a discussion and evaluation. |
| Stage 3 | Stage 3 included the development of functional and spatial solutions on an architectural scale, taking into account the course, legal, and technical requirements as well as universal design, including for people with disabilities. | Stage 1 and 2 and additionally: <br> - floor plans 1:100, 1:200; <br> - characteristic sections 1:100, 1:200; <br> - elevations 1:100, 1:200; <br> - axonometry; <br> - 3D model; <br> - visualizations. | The work was of a design nature, during which various variants of functional and spatial solutions were discussed. The selected variant takes into account the course requirements for the planned development. Analysis and synthesis method: This stage ended with a discussion and evaluation. |
| Stage 4 | Stage 4 consists of the detailed development of material, technology, and construction solutions for a selected part of the building and the calculation of the required design indicators (discussed below). | Stage 1, 2, and 3 and additionally: <br> - construction and building sections; <br> - energy calculations; <br> - LCA analysis; <br> - calculated design indicators with radar diagram. | The work was of a design and analytical nature, during which materials and construction elements of buildings were designed, detailed calculations of design indicators were made, and the project was analyzed in terms of their improvement. The optimal variant was selected. Analysis and synthesis method: This stage ended with a discussion and evaluation. |

In order to perform a detailed evaluation of the project, it was proposed to select 12 project indicators divided into 4 groups. Table 4 summarizes the proposed indicators in conjunction with the EGD (from A to E) and principles of sustainable design (from I to XXVII):

**Table 4.** Proposed design indicators in conjunction with the EGD (from A to E) and principles of sustainable design (from I to XXVII).

| Design Indicator | Value Scale and Assigned Categories of Indicators (from A to D) | EGD (A–E) | SD (I–XXVII) |
|---|---|---|---|
| **Energy:** | | | |
| (E1) Energy demand for heating | A: 0–15 kWh/m$^2$y B: 15–20 kWh/m$^2$y C: 20–30 kWh/m$^2$y D: >30 kWh/m$^2$y | A, B | I, III, IV, V, IX |
| Comment: The expected parameters were selected on the basis of Passive House Institute [50] guidelines. The calculations were made using Sefaira software [51]. | | | |
| (E2) Energy demand for cooling | A: 0–15 kWh/m$^2$y B: 15–20 kWh/m$^2$y C: 20–30 kWh/m$^2$y D: >30 kWh/m$^2$y | A, B | I, III, IV, V, IX |
| Comment: The expected parameters were selected on the basis of Passive House Institute [50] guidelines. The calculations were made using Sefaira software [51]. | | | |
| (E3) Final energy requirement | A: 0–60 kWh/m$^2$y B: 60–80 kWh/m$^2$y C: 80–100 kWh/m$^2$y D: >100 kWh/m$^2$y | A, B | I, III, IV, V, IX |
| Comment: The expected parameters were selected on the basis of Passive House Institute [50] guidelines. The calculations were made using Sefaira software [51]. | | | |
| (E4) Electricity production from a photovoltaic installation | A: 75–100% of the demand B: 50–75% of the demand C: 25–50% of the demand D: <25% of the demand | A, B, E | II, V, IX, XXVII |
| Comment: The expected parameters have been set in relation to the EU targets for the share of renewable energy sources (target: 40% renewable energy by 2030). | | | |
| (E4) Electricity production from a photovoltaic installation | A: 75–100% of the demand B: 50–75% of the demand C: 25–50% of the demand D: <25% of the demand | A, B, E | II, V, IX, XXVII |
| Comment: The expected parameters have been set in relation to the EU targets for the share of renewable energy sources (target: 40% renewable energy by 2030). | | | |
| (E5) The use of elements increasing the energy efficiency of buildings in the design | Heating pumps, ventilation with recuperation, green roofs, solar panels, photovoltaic panels, passive envelope, solar walls, corner ventilation, passive heat gains, shading systems A: lots of items B: lots of items C: few items D: very few items | A, B | I, II, III, IV, V, IX |
| Comment: The proposed elements have an impact on increased energy efficiency [18–21,23–25]. | | | |
| **Environment:** | | | |
| (EN1) Embodied energy | A: 0–5000 MJ/m$^2$ B: 5000–10,000 MJ/m$^2$ C: 10,000–20,000 MJ/m$^2$ D: 20,000–50,000 MJ/m$^2$ | A, B, D | II, VI, XV, XVI, XVII, XVIII, XX, XXI |
| Comment: The expected parameters were developed on the basis of the author's experience in developing LCA analyses. The calculations were made with Athena Impact Estimator software [52]. | | | |

**Table 4.** *Cont.*

| Design Indicator | Value Scale and Assigned Categories of Indicators (from A to D) | EGD (A–E) | SD (I–XXVII) |
|---|---|---|---|
| (EN2) Carbon footprint of materials (LCA) | A: 0–500 kgCO$_2$eq/m$^2$ B: 50–1000 kgCO$_2$eq/m$^2$ C: 1000–2000 kgCO$_2$eq/m$^2$ D: >2000 kgCO$_2$eq/m$^2$ | A, B, D | II, VI, XV, XVI, XVII, XVIII, XX, XXI |
| Comment: The expected parameters were developed on the basis of the author's experience in developing LCA analyses. It should be added that the greater the ratio of the embodied carbon footprint of the materials to the operational carbon footprint is, the more energy-efficient the building is [53]. The calculations were made using Athena Impact Estimator software [52]. | | | |
| (EN3) Use of grey and rain water for flushing toilets and watering the garden | A: 75–100% of the demand B: 50–75% of the demand C: 25–50% of the demand D: <25% of the demand | B, D | V, XXI, XXIII |
| (EN4) Use of mains water for flushing toilets and watering crops | A: <25% of the demand B: 25–50% of the demand C: 50–75% of the demand D: 75–100% of the demand | B, D | V, XXI, XXIII |
| Comment: Great emphasis on extensive rainwater retention systems was adopted in connection with the potential increase in extreme weather phenomena characterized by a greater frequency of droughts and rapid rainfalls [4,54]. | | | |
| (EN5) Vegetable planting | A: >50% of the demand B: 50–40% of the demand C: 40–25% of the demand D: <25% of the demand | C, E | X-XIV, XXV |
| (EN6) Fruit planting | A: >50% of the demand B: 50–40% of the demand C: 40–25% of the demand D: <25% of the demand | C, E | X-XIV, XXV |
| Comment: The calculations were based on the performance indicators of fruits and vegetables in Poland [55,56]. The expected values were based on the author's experience with EGD guidelines. It is expected that at least 25% of the EU's agricultural land will be devoted to organic farming and to significantly increase organic aquaculture by 2030, strengthen local and low-value processing and promote short trade, and reduce climatic and environmental footprints [32]. In addition, the Agricultural European Innovation Partnership (EIP-AGRI) requires the following: To look at the integration of buildings and greenhouses, e.g., by building greenhouses on the roofs of other buildings. CO$_2$ could be used from industrial processes or office buildings. In the case of rooftop greenhouses, the outputs of a building (e.g., wastewater, CO$_2$) could be used as input to the greenhouse and vice versa (e.g., heat from the greenhouse to the building) [36]. | | | |
| (EN7) The use of environmentally friendly elements in the project | EV chargers, parking spaces for bicycles, ecological park, waste segregation and recycling, home gardens and crops, orchards, permaculture, animal farms, hives for bees, permeable surfaces, water retention, use of grey and rain water, wild nature areas. A: lots of items B: lots of items C: few items D: very few items | B, C, D, E | VI, X-XIV, XV, XVIII, XXII, XXIV, XXV |
| Comment: The proposed elements have a positive impact on the environment [18,19,26,27,30,33–36,41–43,45–47,57,58]. | | | |
| **Indoor Climate** | | | |
| (IC1) Lighting (Daylight Factor—DF300) | A: >75% of time of use B: 75–60% of time of use C: 50–40% of time of use D: <40% of time of use | B | VII |

**Table 4.** *Cont.*

| Design Indicator | Value Scale and Assigned Categories of Indicators (from A to D) | EGD (A–E) | SD (I–XXVII) |
|---|---|---|---|
| Comment: DF300 means the percentage of occupied hours where illuminance is at least 300 lux, measured at 0.85 m above the floor plate, and is based on the CEN European Daylight Standard (EN 17037). The calculations were made using Sefaira software [51]. | | | |
| (IC2) The use of elements that increase the comfort of users in the design | Filters in ventilation, humidifiers, night ventilation, corner ventilation, solar towers, zoning heating and cooling, other indoor climate utilities. A: lots of items B: lots of items C: few items D: very few items | B | VII |
| Comment: The proposed elements increase the comfort of users [26,28,29]. | | | |
| **Society** | | | |
| (S1) The use of universal design elements | Relations with surroundings, facilities for the disabled, deaf, and blind. A: lots of items B: lots of items C: few items D: very few items | B | VII |
| Comment: Based on the guidelines of the European Concept for Accessibility (ECA) [59,60]. | | | |
| (S2) The use of elements supporting the creation of social relations | Ecological parks, sensory playgrounds, shared kitchens, small gastronomy spaces, workshops, multi-functional spaces, social gardens, forest kindergartens, ecological education centers, exhibition spaces, sport and fitness utilities, common service spaces, workplaces, shops, fairs, laundries, drying rooms. A: lots of items B: lots of items C: few items D: very few items | B | VII |
| Comment: The proposed elements have an impact on strengthening the feeling of community and social participation [26,28,29]. | | | |

The work uses the Research Through Design (RTG) method, which combines the necessary teaching and interdisciplinary research. The RTD method involves the inclusion of design activities in the research process and the use of design to acquire new quality knowledge, including knowledge about sustainable development. By creating prototypes, patterns, sample projects, and unfinished products, it is possible to explore potential opportunities and innovations without large and risky investments [61]. This method also brought beneficial effects when conducting courses on similar subjects [62].

**7. Results**

On the basis of the completed projects, the values of the sample project indicators obtained in each of the groups were summarized. The summary of the obtained indicators is shown in Table 5.

**Table 5.** The obtained values of some indicators and the categories to which they belong (from A to D).

| Group | Heating Energy | Cooling Energy | Final Energy | Photovoltaics | Embodied Energy | Carbon Footprint | Grey and Rain Water | Mains Water | Vegetables | Fruits | DF300 |
|---|---|---|---|---|---|---|---|---|---|---|---|
| Index (Table 4) | E1 | E2 | E3 | E4 | EN1 | EN2 | EN3 | EN4 | EN5 | EN6 | IC1 |
| Unit | kWh/m$^2$y | kWh/m$^2$y | kWh/m$^2$y | % | MJ/m$^2$ | kgCO$_2$eq/m$^2$ | % | % | % | % | % |
| Group A | 30 | 15 | 64 | 48.5 | n.d. | 209 | 68.9 | 31.1 | 18 | 74 | 75 |
| Group B | 15 | 4.7 | 44 | 100 | 2830 | 204 | 100 | 0 | 87.3 | 54.5 | 75 |
| Group C | 51 | 1.8 | 59 | 45 | 3741 | 173 | 64 | 36 | 63.9 | 52.4 | 60 |
| Group D | 40 | 1.5 | 47 | 100 | 8177 | 413 | 47 | 53 | 112 | 102 | 75 |
| Group E | 29 | 25 | 21 | 76 | 3090 | 420 | 66 | 34 | 100 | 63.7 | 75 |
| Group F | 15 | 15 | 80 | 42 | n.d. | n.d. | 70 | 30 | n.d. | n.d. | n.d. |
| Average | 30.18 | 10.48 | 52.5 | 68.6 | 4459 | 284 | 69.3 | 30.7 | 76.2 | 69.3 | 72 |
| Average Category | C | A | A | B | A | A | B | B | A | A | B |

n.d.—no data.

The best results were obtained for the following indicators: E2 (cooling energy)—average 10.48 kWh/m$^2$y, which means category A; E3 (final energy)—average 52.5 kWh/m$^2$y, which means category A; EN1 (embodied energy)—average 4459.37 MJ/m$^2$, which means category A; EN2 (carbon footprint)—average 283.69 kgCO$_2$eq/m$^2$, which means category A; EN5 (vegetable planting)—average 76.2% of yearly demand, which means category A; EN6 (fruit planting)—average 69.3% of yearly demand, which means category A. The worse category B was obtained by the students for the following indicators: E4 (photovoltaics)—average 68.6% of yearly demand, EN3 (gray and rain water)—average 69.3% of demand for flushing toilets and watering garden crops, EN4 (mains water)—average 30.7% for demand for flushing toilets and watering garden crops, and IC1 (DA300)—average 72% of occupied hours with 300 lux illuminance. The worst result was achieved for the E1 index (heating energy)—an average of 30.18 kWh/m$^2$y—which means category C.

All indicators were presented on a radar chart, selected examples of which, together with illustrative visualizations, are presented in Table 6.

Additionally, during the consultations, each group was encouraged to perform comparative analyses of various variants in order to select a more favorable one for the selected design indicator. In addition, the groups had to perform comparative LCA analyzes by comparing different material variants.

**Table 6.** Projects visualizations, master plans, and radar charts with indicators of each group.

| Group | Visualization and Master Plan | Radar Chart with Design Indicators |
|---|---|---|
| A |  |  |

**Table 6.** *Cont.*

| Group | Visualization and Master Plan | Radar Chart with Design Indicators |
|-------|-------------------------------|-------------------------------------|
| B | | |
| C | | |
| D | | |
| E | | |

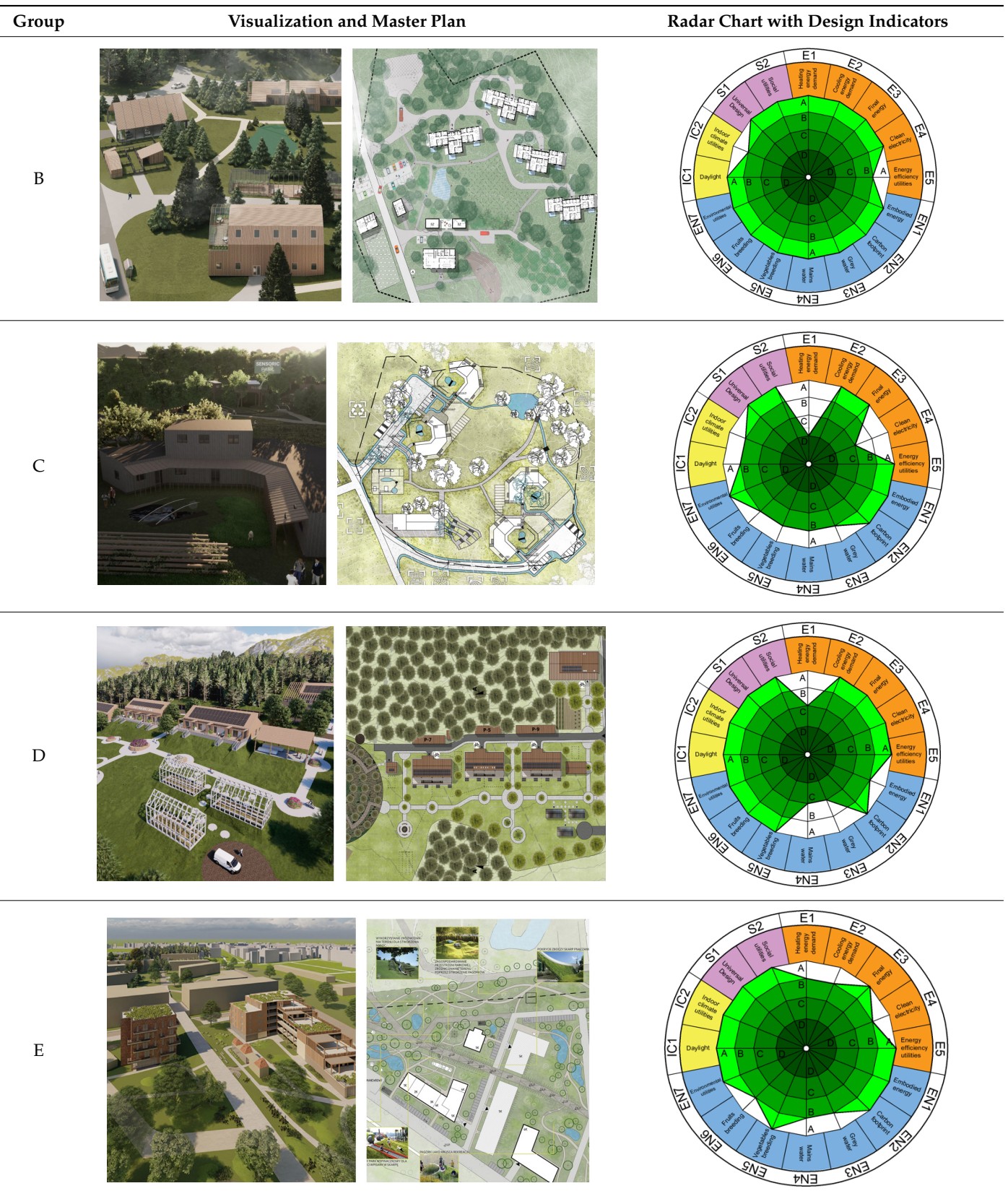

**Table 6.** *Cont.*

| Group | Visualization and Master Plan | Radar Chart with Design Indicators |
|---|---|---|
| F | 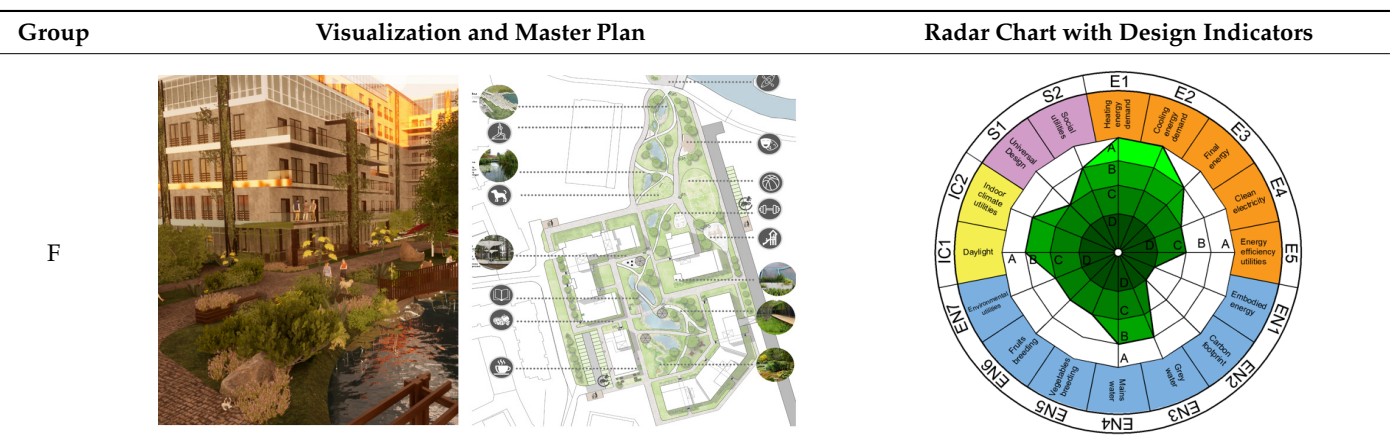 | |

## 8. Course Evaluation

After the end of the course, it was evaluated by means of a questionnaire survey as a didactic and scientific experiment. The aim was to test the effectiveness of the course method and its impact on sustainable design skills and the environmental awareness of the course participants. Moreover, the influence of particular project areas on the possessed skills and the degree of difficulty with their development were examined in detail. The survey included 12 closed questions and one open question; 14 people took part in the survey, which constituted 93% of people participating in the course.

The questions and the results are presented below (Table 7):

**Table 7.** Survey questions with answers.

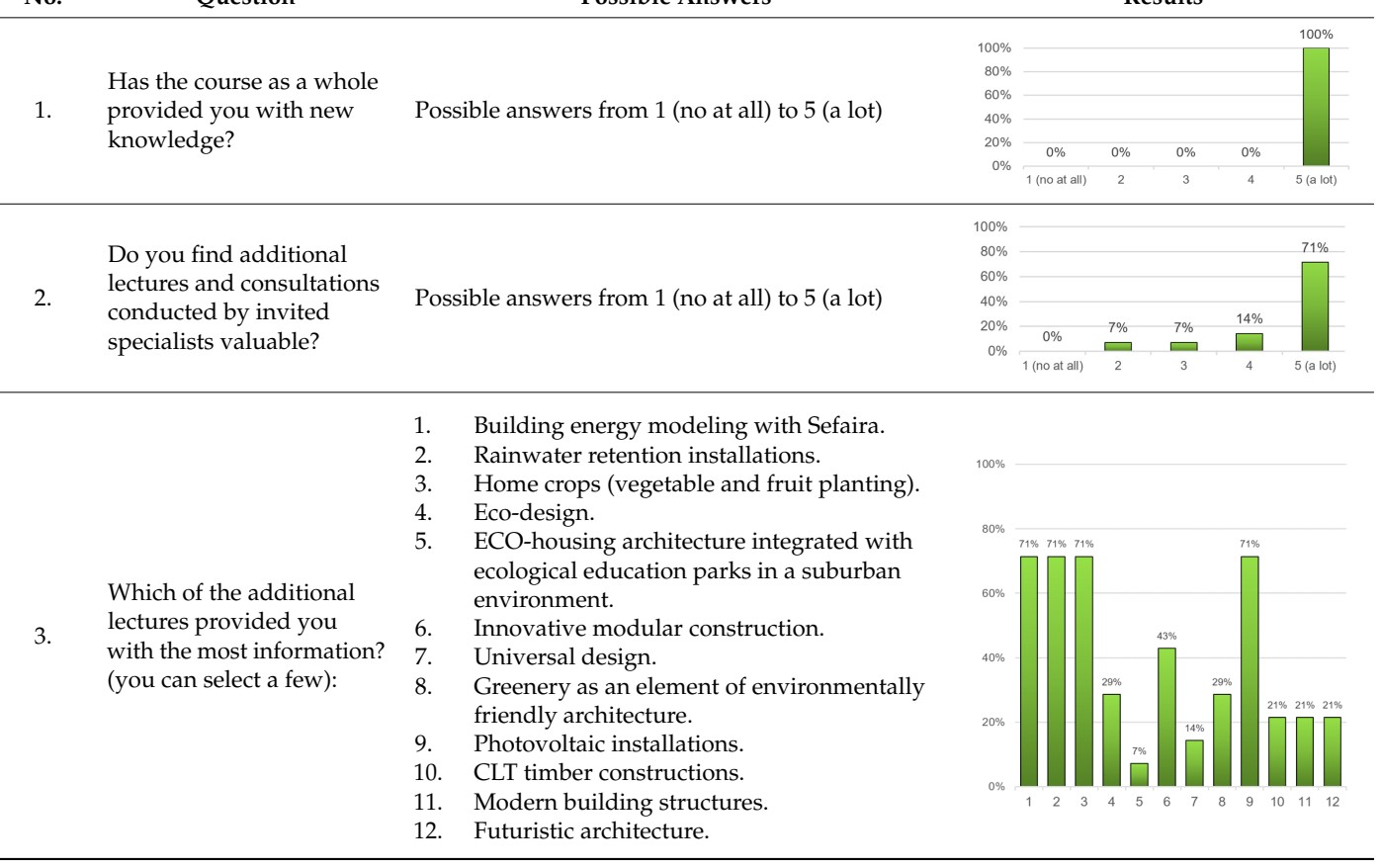

| No. | Question | Possible Answers | Results |
|---|---|---|---|
| 1. | Has the course as a whole provided you with new knowledge? | Possible answers from 1 (no at all) to 5 (a lot) | |
| 2. | Do you find additional lectures and consultations conducted by invited specialists valuable? | Possible answers from 1 (no at all) to 5 (a lot) | |
| 3. | Which of the additional lectures provided you with the most information? (you can select a few): | 1. Building energy modeling with Sefaira. 2. Rainwater retention installations. 3. Home crops (vegetable and fruit planting). 4. Eco-design. 5. ECO-housing architecture integrated with ecological education parks in a suburban environment. 6. Innovative modular construction. 7. Universal design. 8. Greenery as an element of environmentally friendly architecture. 9. Photovoltaic installations. 10. CLT timber constructions. 11. Modern building structures. 12. Futuristic architecture. | |

**Table 7.** *Cont.*

| No. | Question | Possible Answers | Results |
|-----|----------|------------------|---------|
| 4. | In your opinion, did the course take up the issues related to climate change? | Possible answers from 1 (no at all) to 5 (yes definitely) | Bar chart: 1 (no at all) 0%, 2 0%, 3 0%, 4 7%, 5 (yes) 93% |
| 5. | In your opinion, have you gained knowledge on how to design in a climate-responsible way? | Possible answers from 1 (no at all) to 5 (a lot) | Bar chart: 1 (no at all) 0%, 2 0%, 3 0%, 4 21%, 5 (a lot) 79% |
| 6. | How important was it to perform iterative 3D research on a virtual model (Sefaira) to ensure the success of your project (in terms of energy efficiency)? | Possible answers from 1 (not important) to 5 (very important) | Bar chart: 1 (not important) 0%, 2 7%, 3 0%, 4 50%, 5 (very important) 43% |
| 7. | How important was it to perform iterative 3D research on a virtual model (Athena) to ensure the success of your project (in terms of low carbon footprint)? | Possible answers from 1 (not important) to 5 (very important) | Bar chart: 1 (not important) 7%, 2 21%, 3 0%, 4 64%, 5 (very important) 7% |
| 8. | In your opinion, were detailed calculations of project indicators a good idea? | Possible answers from 1 (bad idea) to 5 (very good idea) | Bar chart: 1 (bad idea) 0%, 2 0%, 3 0%, 4 7%, 5 (good idea) 93% |
| 9. | Which project index calculation provided you with the most knowledge and satisfaction? (you can indicate a few) | 1. Heating, cooling, and final energy demand [kWh/m$^2$rok]. 2. Electricity production from PV [% of the demand]. 3. Embodied energy [MJ/m$^2$]. 4. Carbon footprint GWP [kgCO$_2$eq/m$^2$]. 5. Use of grey and rain water [% of demand]. 6. Vegetable production [% of demand]. 7. Fruit production [% of demand]. 8. Lighting DF300 [% of occupied hours where illuminance is at least 300 lux]. | Bar chart: 1 79%, 2 71%, 3 36%, 4 43%, 5 71%, 6 50%, 7 50%, 8 21% |
| 10. | Which design indicator was the most difficult to calculate? (you can indicate a few) | The same answers as in Question 9 | Bar chart: 1 36%, 2 14%, 3 14%, 4 64%, 5 0%, 6 0%, 7 0%, 8 29% |

**Table 7.** *Cont.*

| No. | Question | Possible Answers | Results |
|---|---|---|---|
| 11. | Which indicator do you think you should focus on when designing? (several may be indicated): | The same answers as in Question 9 |  |
| 12. | Would you recommend this course to others? | Possible answers from 1 (no) to 5 (yes definitely) |  |
| 13. | Your opinion about the course? | Some of the answers: "Time well spent—the lectures with discussions at the end were very educational and necessary." "Every student should attend such a course." "The best design classes in which I could participate." "Learning programs and calculations has made the concept of green architecture more specific. The numbers have shown that every building decision matters. From the location and orientation of the body itself, through the design of specific rooms and wall openings, to choosing the best material with a low carbon footprint. The lectures perfectly complemented the knowledge passed on and broadened it." "The possibility of consulting with several experts allowed us to look at the project from different perspectives." | |

The survey showed the usefulness of the presented approach to teaching. All the respondents clearly stated that the course provided them with new knowledge (100%, Question 1), the obtained knowledge would allow them to design in a climate-responsible manner in the future (21% confirmed it and 79% confirmed it strongly, Question 5), and that they would definitely recommend the course to others (93%, Question 12). All respondents agreed (7% confirmed and 93% strongly confirmed, Question 4) that the course addressed the issues of the climate crisis.

Most people (71% strongly confirmed, Question 2) assessed the cooperation (lectures and consultations) with specialists in various fields as valuable, although the assessment of the lectures presented was different: from very good ones related to energy efficiency, rainwater retention, crops, or photovoltaic installations (71% considered that the lecture provided valuable information) to those with lower ratings related to CLT wood, modern structures, or futuristic architecture (only 21% considered that the lecture provided valuable information); lectures on universal design and educational ecological parks were poorly rated (14 and 7%, respectively, Question 3). Additional discussions showed that the lectures that conveyed specific knowledge based on numerical data and calculation methods based on presented formulas and given examples were better assessed. Thanks to this, the knowledge from the lecture could be implemented in course projects.

The course participants received the idea of calculating the design indicators presented on a radar chart very well (93% strongly stated that it was a good idea, while 7% said it was a good idea; Question 8) despite the fact that it required more work from them and involved mastering new skills, often from specializations other than architecture. It should be added that most design courses do not offer such knowledge and do not require precise calculations (based on the author's interview with the authors of other courses). It was assessed that calculating at least half of the indicators brought a lot of satisfaction and

knowledge (average 53%, Question 9), with the most satisfactory being the calculation of energy efficiency indicators (79%), electricity production from PVs (71%), and use of gray and rain water (71%, Question 9). According to the indications of Question 10, these were also easy to calculate indicators (only 36%, 14%, and 0%, respectively, had difficulties in calculating a given indicator), and their significance for the design process was also assessed differently (79%, 50%, 7%, respectively).

Other interesting conclusions show that, for example:

- calculations of energy efficiency were considered important (79%, Question 11), not difficult to calculate (36% had difficulties, Question 10), and satisfactory (79%, Question 9), which suggests their strengthening in the future.
- calculations of the current from PV installations were considered satisfactory (71%, Question 9) and very easy (14.3% had difficulties, Question 10) but not important for everyone (50%, Question 11).
- calculating the embodied carbon footprint was considered the most difficult (64% had difficulties, Question 10) and, at the same time, one of the most important in design (79%, Question 9), which suggests the need to develop tools to support its teaching and calculation.
- calculating the possibility of local food production was assessed as satisfactory by half of the respondents (50%, Question 9) and very easy to perform (0%, Question 10) but relatively irrelevant when designing (7%, Question 11).

## 9. Conclusions

The work demonstrates the relationship between the historical Bauhaus art school, the Baukultur movement concept derived from it, the assumptions of the NEB, and the assumptions of sustainable design. The method of teaching the design course *Environmentally Friendly Housing Architecture* has been indicated, as well as the method of evaluating the work performed and the course itself. Certain assumptions, e.g., an innovative approach to design, group work, cooperation with specialists from many fields, learning the practical skills of calculating the proposed design indicators, and striving to improve them, have brought a positive effect. The completed questionnaire survey showed the correctness of the assumptions, methods, and project evaluation criteria used, as well as the high didactic value of the course. The need for additional work in the form of calculating project indicators was well received because it brought a lot of satisfaction and new knowledge. Particularly valuable is the information that the ability to calculate the embodied carbon footprint of a building is a difficult but desirable skill; this was assessed as very important in the design process. This is particularly important because the Committee on Environment, Public Health, and Food Safety has adopted its Report On the New EU Circular Economy Action Plan, where it calls for science-based and binding 2030 EU targets for materials use and consumption footprints, covering the whole lifecycle of each product category placed on the EU market. To this end, they urge the Commission to introduce in 2021 harmonized, comparable, and uniform circularity indicators for material and consumption footprints [63]. Additionally, because the research shows that according to the construction sector, architectural specialization is the most responsible factor for the decarbonization of construction [64], it is necessary to quickly develop a uniform methodology and create appropriate tools that can serve as the basis for the development of teaching plans at architectural universities.

The work uses the aforementioned RTG methodology, thanks to which it is possible to search for optimal building parameters using a digital model in terms of the expected values of design indicators. In a complex and multi-criteria environment, the multi-criteria optimization of designed buildings, carried out in cooperation with many specialists, brought the participants of the course a lot of knowledge and satisfaction, with the buildings designed in an environmentally friendly way.

Taking into account the above assumptions, the positive reception of the course, and the results achieved through optimization of the adopted solutions, the teaching method

developed for the needs of the design course *Environmentally Friendly Housing Architecture* course can be used by other educators.

## 10. Future Works

In the future, the author intends to devote more time to the modernization of existing buildings, which will correspond to the Renovation Wave program implemented by the European Commission under the EGD. The proposed 'build less' approach, maximizing the use and exploitation of existing assets, could enable an 80% reduction in carbon emissions, as opposed to the 'build smart' (50% reduction) and 'build efficient' (20% reduction) approaches, which are insufficient from the point of view of the strategy to achieve the goal of climate neutrality [65].

Additionally, after the European Commission announces the winners of the NEB competition, the distinguished ideas and projects will be analyzed and implemented.

**Funding:** This research received no external funding.

**Institutional Review Board Statement:** Not applicable.

**Informed Consent Statement:** Not applicable.

**Data Availability Statement:** The survey presented in this study are openly available at https://docs.google.com/forms/d/12bFI4kaa3Je2YiXTKlBDWjLpansDQUbcIeknOg9Q4t8/viewanalytics (accessed on 24 August 2021).

**Acknowledgments:** The author would like to thank everyone without whom the writing of this article would not be possible. In particular, author would like to thank Woytek Kujawski, who provided help during the design course: Environmental Friendly Housing Architecture and transferred practical and theoretical knowledge based on his sustainable design experience in Canada and abroad. He also supervised the cooperation of students with the owner of the property in Wakefield (Quebec), Christopher Minnes. Author would also like to thank Christopher for offering his property for the eco-design of a small community, communicating his expectations and providing ideas and valuable comments for potential development during the course. Author would also like to thank all students who took part in the course and the survey.

**Conflicts of Interest:** The authors declare no conflict of interest.

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
