# Peer review of "Implementation of the New European Bauhaus Principles as a Context for Teaching Sustainable Architecture"

_sustainability, doi:10.3390/su131910715_

Round 1

Reviewer 1 Report

 I found the subject interesting and I appreciate the opportunity to read it

Please note that in line 63 the word “Bauhaus” are misspelled.

I found that naming the criteria “use of rain water as gray water” (line 19) to express the amount of water reduced do to the use of rain water, it is confusing.

I am afraid I do not found clear what is “The course” mentioned in line 24, since it was not presented before.

I found that some of the “Radar charts” on Table 6 are difficult to read, if possible present all like the first one from group A.

Location and climate play a very important role to predict the energy needs for heating and cooling but is also important to be taking in the account for “design of multi-family residential buildings” (line 209). My suggestion is to include the locations that were considered.

Finally I think that conclusions and future developments should be separate. The ideas in line 424 may benefit from a separate section.  

In the overall I appreciate work developed and the paper.

Reviewer 2 Report

The paper describes the implementation of the new European Bauhaus principles as a context for teaching sustainable architecture. In the manuscript the statistical summary of the achieved values for individual indicators, the progress achieved and, exemplary design solutions, and assessment of methodology was presented. The manuscript is clear, well-organized and interesting. Only some parts of manuscript need improvement i.e.

  • Please remove citation from abstract.
  • Line 24 – Please change “and assesment of methodology” to “,and assessment of methodology.
  • Table 5, and Line 321 – change “*” to “”
  • Table 6 - the font in the pie chart is too small. This makes the drawing unreadable.
  • Table 7 - Number 3 is missing from the table.
